# Mindfulness-Based Interventions for Undergraduate Nursing Students in a University Setting: A Narrative Review

**DOI:** 10.3390/healthcare9111493

**Published:** 2021-11-02

**Authors:** Clare McVeigh, Lindsay Ace, Chantal F. Ski, Claire Carswell, Stephanie Burton, Soham Rej, Helen Noble

**Affiliations:** 1School of Nursing and Midwifery, Queen’s University Belfast, Belfast BT7 1NN, UK; lace02@qub.ac.uk (L.A.); c.carswell@qub.ac.uk (C.C.); sburton06@qub.ac.uk (S.B.); helen.noble@qub.ac.uk (H.N.); 2Integrated Care Academy, University of Suffolk, Ipswich IP4 1QJ, UK; c.ski@uos.ac.uk; 3Department of Health Sciences, University of York, York YO10 5DD, UK; 4McGill Meditation and Mind-Body Medicine Research Clinic (MMMM-RC) and Geri-PARTy Research Group, Jewish General Hospital/Lady Davis Institute, Montreal, QC H3T 1E2, Canada; soham.rej@gmail.com; 5Department of Psychiatry, McGill University, Montreal, QC H3A 0GA, Canada

**Keywords:** mindfulness, meditation, nurse education, nurses, nursing students, healthcare education, stress

## Abstract

(1) Introduction: Undergraduate (UG) nursing students are vulnerable to stress throughout their education, known to result in burnout, with high attrition rates of up to 33%. There is a growing body of evidence to suggest that mindfulness-based interventions are effective for the management of anxiety, depression and wellbeing, thereby reducing stress in healthcare provider populations. The aim of this narrative review was to synthesize and provide a critical overview of the current evidence in relation to mindfulness-based interventions for UG nursing students in a university setting. (2) Methods: A review of the literature was conducted in March 2020 and updated in May 2021, utilising the databases CINAHL, Medline and PsycINFO. (3) Results: Fifteen studies were included in the review, with three common themes identified: (i) the positive impact of mindfulness on holistic wellbeing, (ii) mindfulness-based techniques as a positive coping mechanism within academic and clinical practice, and (iii) approaches to the delivery of mindfulness-based interventions. (4) Conclusions: Mindfulness-based interventions are effective strategies for the management of stress, development of self-awareness and enhanced academic and clinical performance in undergraduate nursing students. No ideal approach to delivery or duration of these interventions was evident from the literature. Best practice in relation to delivery of mindfulness-based interventions for nursing students is recommended for future studies.

## 1. Introduction

Nursing students are particularly vulnerable to high levels of stress during their education. Common stressors include clinical workload, academic performance concerns, lack of monetary resources, lack of support during clinical placements, and the death and suffering of patients [1]. The associated psychological stress can have significant negative effects on nursing students, often leading to burnout and a desire to leave the profession [2]. A systematic review highlighted that nurses can experience higher levels of stress-related burnout, in comparison with other healthcare professionals [3]. Burnout can have a negative impact on the bio-psychosocial wellbeing of nursing students. Additionally, stress and burnout can lead to higher staff turnover amongst newly qualified nurses [4]. A recent report in 2018 from Health Education England [5] highlighted the concerning rates of student nurse attrition, with a dropout rate of 33.4% for the academic years 2013–2014 and 2014–2015. In addition, 41% of surveyed students had considered leaving their programme of study [5]. The current COVID-19 global pandemic has additionally impacted the psychosocial wellbeing of undergraduate (UG) nursing students [6,7]. It is therefore important to consider the psychological wellbeing of UG nursing students to prepare them for future healthcare challenges.

Mindfulness-based interventions (MBIs) can positively impact stress, anxiety and general wellbeing in healthcare students undertaking clinical training [8,9]. Mindfulness is the psychological process of bringing one’s attention, in a judgement-free manner, to what is being experienced in the present moment [10]. It uses formal meditation techniques, such as the body scan, mindful movement and sitting meditation, and informal daily meditation practice to develop an awareness of one’s experiences [11]. Events are viewed within an attitudinal framework of kindness and curiosity, offering the potential for development of new insights into how one relates to the present moment [12].

Mindfulness-based stress reduction (MBSR) is an evidence-based psycho-educational eight-session programme pioneered by Kabat-Zinn to incorporate mindfulness meditation practice into an accessible format [13]. Originally developed for the treatment of patients with chronic pain, there is now a body of evidence identifying a range of beneficial outcomes, such as decreased fatigue in patients with cancer; improvements in insomnia, anxiety, depression and wellbeing; and improved memory and cognitive functioning in older adults [14,15,16]. Mindfulness-based stress reduction has also been adapted to a wide variety of contexts and highly stressful environments, such as prisons, corporate settings and medical schools [17,18,19]. In addition, MBSR is frequently delivered over varying lengths of time [20], and other interventions such as mindfulness-based cognitive therapy (MBCT) [21] have since been developed with the principles of mindfulness practice embedded within them. 

The overall aim of this narrative review was to explore the effectiveness of mindfulness-based meditation interventions for UG nursing students in a university setting.

## 2. Materials and Methods

### 2.1. Search Strategy

A review of the literature was conducted in March 2020 and updated as new research was published until May 2021. The search strategy is described in Figure 1. Databases, including CINAHL, Medline and PsycINFO, were accessed online to identify relevant studies and primary research. These electronic databases were chosen as they covered a wide range of international psychology, medical, nursing and social care literature [22]. The relevant databases were searched using the identified key terms: “healthcare student”, “nursing student”, “baccalaureate nursing student”, “mindfulness” and “meditation”. The acronym PEO, which stands for population, exposure and outcome [23], was used to guide the development of key terms.

In total, 173 articles were identified through the search. Of these, 35 duplicates were removed. Inclusion and exclusion criteria were developed to ensure that only studies specifically related to the topic area being explored were included in the review of the literature (Table 1). After application of the inclusion criteria, 123 articles were excluded. Fifteen studies were included in the narrative review: 11 quantitative [24,25,26,27,28,29,30,31,32,33,34], 3 qualitative [35,36,37] and 1 mixed-methods study [38]. Within the quantitative studies, there were quasi-experimental studies (n = 3) [24,28,34], randomised controlled trials (RCTs) (n = 5) [25,26,27,29,31], an interventional study (n = 1) [30] and a survey study (n = 1) [33].

The methodological quality of the studies was reviewed by a member of the research team (LA) using the Critical Appraisal Skills Programme (CASP) [39] critical appraisal tools, LA and CMV also discussed and appraised the literature together. The CASP tools provided an evidence-based approach to assessing the quality, quantity and consistency of the research designs applied by the research included in the review [40]. The characteristics of the studies included in the literature review, and their outcomes measures [41,42,43,44,45,46,47,48,49,50,51,52,53,54,55,56,57,58,59], are contained in Appendix A.

### 2.2. Narrative Synthesis

A narrative synthesis was undertaken, using thematic analysis to synthesize the findings of the studies included within the review [60]. This approach has been recommended for reviews which include studies with a range of research designs [60]. The use of themes was chosen by the researcher to appropriately identify the most commonly occurring issues in the literature [22]. Using themes allowed the researchers to show the relationships among the studies included in the review and facilitated the ongoing critical analysis of the studies, until a comprehensive understanding of the relevant phenomena was reached [61]. Descriptive themes were initially assigned to the findings from the individual studies and then grouped together to display the overarching key themes [62]. The initial narrative synthesis was conducted by LA, with further analytical input from CMV.

## 3. Results

The studies took place across several countries including USA (n = 7), Turkey (n = 2), China (n = 1), South Korea (n = 1), Australia (n = 1), Canada (n = 1), Sweden (n = 1) and The Netherlands (n = 1). The sample size of participants ranged from 5 to 201, and all studies included undergraduate nursing students. Through the narrative synthesis, the 15 studies generated three key themes: the positive impact of mindfulness on holistic wellbeing, mindfulness-based techniques as a positive coping mechanism within academic and clinical practice; and approaches to the delivery of mindfulness-based interventions.

### 3.1. Positive Impact of Mindfulness on Holistic Wellbeing

Mindfulness-based interventions reduced stress [25,27,29,31,32,34], anxiety [24,26,29,31,34] and depression [31], whilst increasing mindfulness [25,27,30,31,38] and awareness [28] in UG nursing students. Changes in stress levels were also maintained in studies which included a follow-up period, with persistent effects reported at 3 months [27], 4 months [34] and 24 weeks [32]. Koren [28] did not report a significant change in perceived stress following mindfulness training; however, the sample size for this study was relatively small (n = 13), and the intervention only involved 10 min of weekly mindfulness meditation over a 6-week period. Both Karaca and Şişman [27] and Song and Lindquist [31] reported a statistically significant increase in post-test mindfulness scores when compared with a control group following a MBSR intervention; however, the positive effects on mindfulness scores were not maintained at the end of the 3-month follow-up period [27]. Significant post-test differences in mindfulness scores between the intervention and control groups were reported following delivery of an online mindfulness meditation programme [25]. One randomised controlled trial that compared the effects of mindfulness meditation with biofeedback and a control group reported that whilst biofeedback lowered anxiety levels, it had no significant effect on stress levels, whereas significant improvements were seen in both stress and anxiety levels for the mindfulness group [29]. One study measured physiological responses to a mindfulness intervention, finding an average systolic blood pressure reduction of 2.2 mmHg following seven consecutive days of mindfulness meditation [26].

Mindfulness-based interventions were utilised to not only bring attention to the present moment in a mindful manner but also to develop self-awareness. Enhanced self-awareness enabled identification of life stressors and proactive management of certain situations through self-regulation. This was frequently described as a specific skill or tool which could be used in particularly stressful or demanding situations [27,35,36,37]. The effectiveness of a 4-h once-weekly mind–body skills intervention in Swedish nursing students (n = 48) was conducted by van Vliet et al. [27] over a 5-week period. This study reported greater self-awareness, with increased insight into personality and preferences. Following an analysis of 91 self-reflection reports and a focus group with nursing students (n = 3), Niessen and Jacobs [38] concluded that participants reported positive effects on their self-insight following a 4-week pilot training course on mindfulness.

Mindfulness-based techniques were commonly described to manage disruptive thoughts or emotions, maintaining a state of equanimity and deciding to view the situation from a different perspective [35,37]. The results highlighted the positive impact that mindfulness-based interventions had on an individual’s self-efficacy [35,37]. Self-efficacy can be described as an individual’s belief in their ability to successfully exert control over their behaviour and, according to self-efficacy theory, is a strong determinant of whether an individual will engage in that behaviour [63]. Two studies reported an understanding that feelings and emotions were not external to them, but rather something which they could choose to regulate, replacing negative self-viewpoints with a more compassionate and accepting attitude [35,37].

### 3.2. Mindfulness-Based Techniques as a Positive Coping Mechanism within Academic and Clinical Practice

A further theme which emerged from this review was the use of mindfulness-based techniques as a positive coping mechanism. Van Vliet et al. [37] reported a reduction in negative coping strategies for stress management and a subsequent ability to more effectively manage workload pressures. Karaca and Şişman [27] reported greater use of self-confident and optimistic approaches to coping with stress, and less use of the helpless approach amongst UG nursing students following a 12-week MBSR intervention (n = 114). Following mindfulness-based interventions, several studies demonstrated an insight into how mindfulness and self-efficacy improved academic and clinical performance [25,33,36,38]. The reported benefits of applying mindfulness skills, acquired during interventions, included the ability to be more focused, patient and observing, and improved executive attention [25,33,36,38].

### 3.3. Approaches to Delivering Mindfulness-Based Interventions

Overall, 47% (n = 7) of studies referred to mindfulness meditation [25,26,28,29,30,33,38], 33% (n = 5) involved an MBSR intervention [24,27,31,32,35] and 20% (n = 3) involved other forms of mindfulness-based interventions, including a stress management and mindfulness programme [36], mindfulness based cognitive therapy [34] and a mind–body skills course [37]. Spadaro and Hunter [32] delivered MBSR in an online format, whilst Marthiensen et al. [35] implemented the Brief MBSR programme. Ratanasiripong et al. [29] had two interventional groups within their RCT: whilst one group took part in mindfulness meditation, the other group adapted biofeedback. Biofeedback is a non-invasive intervention in which a mechanical device is utilised to measure a participant’s physiological responses, allowing them to learn how to modify these responses through control of their internal process [64]. Vankuiken et al. [33] delivered mindful movement strategies to their study population, in addition to mindfulness meditation. The intervention period varied across all studies, ranging from 1 to 12 weeks, with each session lasting anything from 10 min to 8 h. The studies within this review used a diverse range of mindfulness-based interventions, with notable heterogeneity in the duration of sessions, the total number of sessions and the method of course delivery.

Across all studies included within the review, data related to programme adherence was only reported by two studies [30,36]. In one study, 60% (n = 27) of participants attended all eight sessions of mindfulness meditation [30], whilst in another, only one participant attended all 7 weeks of the stress management and mindfulness programme [36]. Despite the low attendance reported by van der Riet et al., focus group data suggested that participants found improvements in their self-awareness and focus on academic studies [36]. Reported challenges to adherence included finding time to prioritise mindfulness practice and other stressors such as academic workload [30].

## 4. Discussion

Knowing oneself is an essential feature of compassionate nursing care [65]. Accepting one’s limitations can be difficult for nurses; however, it can be argued that it is a necessary strategy for survival [66]. Within this review, it was highlighted that mindfulness-based interventions improved the holistic wellbeing of UG nursing students due to reductions in stress [25,27,29,31,32,34], anxiety [24,26,29,31,34] and depression [31]. Mindfulness was even shown to improve physiological health through a reduction in systolic blood pressure [26]. Characteristics of the studies included in the literature review are contained in Appendix A. These results are consistent with other studies which show that mindfulness-based interventions can positively impact stress, anxiety and mindfulness in college [67], UG health and social care [12] and postgraduate nursing [9] students.

By using mindfulness-based skills, UG nursing students can develop a repository of positive coping mechanisms that will increase their capacity to navigate the more stressful aspects of academic and clinical practice. The reported benefits of applying mindfulness skills acquired during interventions included the ability to be more focused, patient and observing, and improved executive attention [25,33,36,38]. These results are consistent with other studies on the effects of mindfulness-based interventions on undergraduate students, which demonstrated increased self-efficacy and use of positive coping strategies [68,69,70,71], with one study reporting a sustained effect 6 years post-intervention [72]. Chen et al. [73] argued that higher levels of self-efficacy in nursing students is a significant predictor of academic achievement and is positively related to clinical competence and performance. Several studies have demonstrated the direct relationship between mindfulness and self-efficacy [74], and the reported link between high self-efficacy and lower stress levels [75].

Previous research has additionally explored the benefits of mindfulness-based interventions for nursing students’ clinical practice, reporting an increased awareness of the connection between mind and body, the importance of considering the patient’s holistic needs, and an ability to acknowledge and accept their own capabilities and limitations as a caregiver [30,33,35,36,37]. Mindfulness-based interventions can also result in increased empathy and communication skills, and greater ability to be present with patients [30,33,35,36,37], enhancing the development of nursing students’ self-awareness. Sanko et al. reported that nursing students not only demonstrated increased mindfulness but also enhanced ethical decision-making when caring for patients [30]. Van Kuiken et al. highlighted that mindfulness interventions additionally helped nursing students in the classroom environment through enabling participants to feel relaxed, calm and more focused when learning in the university setting [33]. Through enhancing internal coping mechanisms [35] and positively impacting sleep, concentration and clarity [36], mindfulness interventions demonstrated a positive impact on how nursing students managed stress. Van der Riet et al.’s findings additionally displayed a reduction in negative cognitions amongst participants [36]. Mindfulness interventions can also improve nursing students’ communication with their patients through enhancement of their ability to be more present and to develop deeper connections with the people in their care [37].

However, there is debate within the literature regarding the delivery method of mindfulness-based interventions. The traditional MBSR programme developed by Kabat-Zinn et al. [13] is typically administered as 2.5 h-long group sessions over 8 weeks, with one 6-h retreat and daily home practice. Whilst there is good evidence to support the effectiveness of the traditional programme in both clinical and non-clinical populations [13], one of the major barriers to mindfulness practice is the time commitment required from participants, resulting in attrition [76]. Despite variations in the duration and type of mindfulness programme utilised in the studies included in the present review, all studies reported a perceived positive impact following the intervention.

Three of the studies within this review delivered their mindfulness intervention online [25,30,32]. These studies reported decreased anxiety and stress, and increased mindfulness, with decreased stress levels maintained at 24 weeks [32] and 4 weeks [25] of follow-up. Previous pilot work by the authors highlighted that 71.6% (n = 149) of UG nursing students perceived that a mobile application would enhance their engagement with mindfulness practice [77]. Delivering mindfulness-based interventions online is convenient, and the literature suggests that it may offer comparable benefits to traditional face-to-face training, showing improvements in empathy, stress, mindfulness and resilience [78,79]. The results from the wider literature also show promising results in student populations using guided mindfulness-based training delivered via smartphone apps or audio CDs, with positive effects noted in stress, anxiety, depression, wellbeing, resilience and mindfulness [80,81,82,83,84]. However, participants in the studies included within the present review reported feeling a sense of closeness with their peers and being able to relate their own lives to others’ more clearly through face-to-face group sessions [33,37]. What is evident from this review of the literature is that there is currently no standardised approach to mindfulness-based interventions for nursing students in terms of programme duration and method of delivery.

This review must be interpreted in the context of several limitations. The relatively small sample sizes and the mainly female populations in many of the studies included may limit the generalisability and representativeness of the results. Additionally, all studies but one relied on convenience sampling, which may have introduced bias and again affected representativeness of the sample. Several of the studies did not include a long-term follow-up, so it is not possible to confirm if the effects of the intervention were sustained. All reviews are potentially open to publication bias, as other relevant studies with negative results may have been withheld from publication. Lastly, the review only included those studies published in English, and this also may have excluded other relevant studies.

## 5. Conclusions

Mindfulness-based interventions have the ability to improve the levels of stress [25,27,29,31,32,34] and anxiety [24,26,29,31,34] experienced by UG nursing students, whilst also increasing their own mindfulness practice. The impact of mindfulness-based interventions was also examined in the context of self-efficacy and self-awareness. The results suggested that participants were able to apply mindfulness skills as a positive coping mechanism for effective stress management [25,27,29,31,32,34]. Viewing oneself in a mindful, non-judgmental manner also allowed participants to develop their self-awareness [28] and acknowledge their limitations, resulting in increased academic focus and an enhanced capacity for empathetic, patient-centred care [24].

Whilst exposure to various academic and clinical stressors may be an inevitable aspect of the nursing student experience, there is scope for mindfulness-based interventions, which target the management of stress and anxiety, to be offered within the nursing curriculum [77]. Substantial variation in the intervention durations and methods of delivery were observed, with no ideal approach identified. Completing online training was also associated with positive benefits. As academic and clinical demands increase, nursing students may be unwilling to commit to time-consuming programmes; therefore, interventions delivered online or via a smartphone app may be more desirable. However, the use of online or digital modalities to deliver mindfulness-based interventions is an area that could benefit from further investigation. Alongside this, future research should explore approaches to engaging UG nursing students in mindfulness practice that will deliver and sustain long-term effects.

## Figures and Tables

**Figure 1 healthcare-09-01493-f001:**
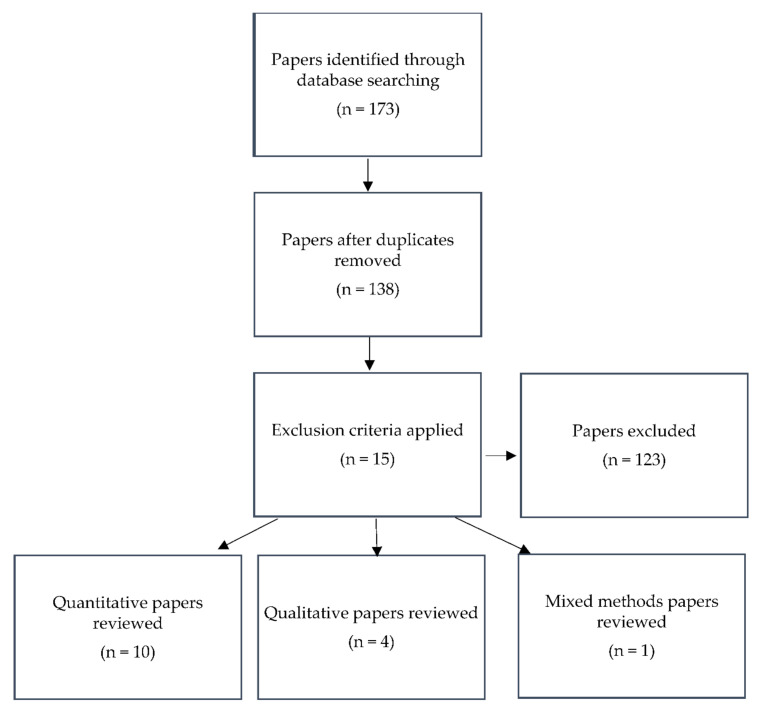
Search strategy flow diagram.

**Table 1 healthcare-09-01493-t001:** Table outlining inclusion and exclusion criteria.

Inclusion criteria
Studies that specifically referred to a mindfulness-based intervention Full-text studies written in English Participants included undergraduate nursing students
Exclusion criteria
Published more than 10 years previous to the date of the initial database search Did not report nurse data separately from other healthcare professionals Studies evaluating the effects of mindfulness interventions on participants diagnosed with a mental health condition such as depression or anxiety disorder

## Data Availability

Not applicable.

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
