# Peer review of "Mindfulness-Based Interventions for Undergraduate Nursing Students in a University Setting: A Narrative Review"

_healthcare, 2021, doi:10.3390/healthcare9111493_

Round 1

Reviewer 1 Report

Dear Authors

Thank you for the opportunity to review your article.

Brief summary: This is a narrative review that aimed to synthesize and provide a critical overview of the current evidence in relation to mindfulness-based interventions for UG nursing students in a university setting. Mindfulness-based interventions are effective strategies for the management of stress and anxiety, development of self-awareness and enhanced academic and clinical performance in undergraduate nursing students.

Areas of strength

The references included are relevant for the subject under study show 37/85(44%) references from the last 5 years. There is strong concordance between the and the methods used. The description of the methodology was made in a clear and adequate way. The discussion correlates with the presented data and takes the published literature into account. The manuscript reports several limitations and important practical implications to undergraduate nursing students. Congratulations to the authors.

Weakness:

Page 3, line 83 – in the box, missing (n=1) in Mixed Methods papers reviewed

Page 8, line 296 – 467. Review the referencing style. Put the year in bold and the journal abbreviations.

Author Response

Thank you for taking the time to provide feedback on our submission. Comments have been addressed as follows:

Page 3, line 83 – in the box, missing (n=1) in Mixed Methods papers reviewed

  • Thank you, this has now been inserted (Page 3, line 124)

Page 8, line 296 – 467. Review the referencing style. Put the year in bold and the journal abbreviations.

  • Thank you, this has now been addressed throughout the reference list

Reviewer 2 Report

(Conclusion) About mindfulness programs available online, it has been argued that the time consuming of the program is not compliant for students. However, this is at odds with the discussion of making the mindfulness program part of the curriculum. In the context of this study, nursing students are exhausted by the curriculum and have mental weaknesses. While it is understandable that mindfulness programs are still being explored, the easy association of online programs as effective, because they are time-consuming and may be avoided by students, raises the tricky question of whether mindfulness programs are more time-consuming than effective for students. It seems to avoid the tricky issue of mindfulness programs being more time-consuming than effective for students. I think it would be better to advance the argument that mindfulness programs, even if they are time-consuming programs (everything is time-consuming), are meaningful to nursing students and should be recommended as a standard curriculum.

The end of the conclusion is in the context of recommending online. I do not deny the convenience of online or its affinity for students. However, as stated in P7L251, the effectiveness of online is still debatable and should not be written in a strong tone because it may mislead the reader.

Please elaborate on how the mindfulness program hypothetically works specifically with nursing students, as stated on P7L274. In the part on P6L224-230, the authors mentioned the results of the review. However, there is little explanation for that mechanism. The academic difficulties faced by nursing students and their emotional control over patients are very important educational issues. I would like to see a discussion of the mechanisms by which mindfulness programs can be effective not only in calming the mind but also in addressing these unique issues in learning nursing.

This is related to the above, though, in the discussion (P6L208), it was stated that the effects of the mindfulness program on nursing students were consistent with the effects on other students and nurses after graduation, but this is not the only aspect of advantage in this study. In this study, I would like to know the program and its unique association with nursing students, and I would like to see a deeper discussion of this unique association. I would like you to explain the specificity of the mindfulness program in terms of its effectiveness indicators in the details if any.

Author Response

Thank you for taking the time to review our submission. Comments have been accommodated ads follows: 

(Conclusion) About mindfulness programs available online, it has been argued that the time consuming of the program is not compliant for students. However, this is at odds with the discussion of making the mindfulness program part of the curriculum. In the context of this study, nursing students are exhausted by the curriculum and have mental weaknesses. While it is understandable that mindfulness programs are still being explored, the easy association of online programs as effective, because they are time-consuming and may be avoided by students, raises the tricky question of whether mindfulness programs are more time-consuming than effective for students.

It seems to avoid the tricky issue of mindfulness programs being more time-consuming than effective for students. I think it would be better to advance the argument that mindfulness programs, even if they are time-consuming programs (everything is time-consuming), are meaningful to nursing students and should be recommended as a standard curriculum. The end of the conclusion is in the context of recommending online. I do not deny the convenience of online or its affinity for students. However, as stated in P7L251, the effectiveness of online is still debatable and should not be written in a strong tone because it may mislead the reader.

  • 7, line 333 has been adjusted to say that mindfulness can be offered within the nursing curriculum. Thus not implying it should be mandatory.

  • 7, line 337 – 338 changed to “As academic and clinical demands increase, nursing students may be unwilling to commit to time-consuming programmes, therefore interventions delivered online or via a smartphone app may be more desirable. However, the use of online or digital modalities to deliver mindfulness based interventions is an area that could benefit from further investigation.” This was to ensure wording was not as strong.

Please elaborate on how the mindfulness program hypothetically works specifically with nursing students, as stated on P7L274. In the part on P6L224-230, the authors mentioned the results of the review. However, there is little explanation for that mechanism. The academic difficulties faced by nursing students and their emotional control over patients are very important educational issues. I would like to see a discussion of the mechanisms by which mindfulness programs can be effective not only in calming the mind but also in addressing these unique issues in learning nursing.

  • Further discussion of the specific findings of these studies has been added (P6, lines 274 – 286)

This is related to the above, though, in the discussion (P6L208), it was stated that the effects of the mindfulness program on nursing students were consistent with the effects on other students and nurses after graduation, but this is not the only aspect of advantage in this study. In this study, I would like to know the program and its unique association with nursing students, and I would like to see a deeper discussion of this unique association. I would like you to explain the specificity of the mindfulness program in terms of its effectiveness indicators in the details if any.

  • Thank you, the following line has been added to this section (P6 lines 250 -251) “Characteristics of the studies included in the literature review are contained in Supplementary Material 1.”